# Myeloperoxidase and Lysozymes as a Pivotal Hallmark of Immunity Status in Rabbits

**DOI:** 10.3390/ani10091581

**Published:** 2020-09-04

**Authors:** Rafał Hrynkiewicz, Dominika Bębnowska, Paulina Niedźwiedzka-Rystwej

**Affiliations:** Institute of Biology, University of Szczecin, Felczaka 3c, 71-412 Szczecin, Poland; rafal.hrynkiewicz@gmail.com (R.H.); bebnowska.d@wp.pl (D.B.)

**Keywords:** myeloperoxidase, lysozyme, rabbits, viral infection, rabbit hemorrhagic disease

## Abstract

**Simple Summary:**

Rabbit breeding is a very important element in the context of broadly understood industrial breeding, as rabbits are one of the main and most frequently chosen economic directions. Effective rabbit breeding, however, requires full control over the health of these animals, which is particularly related to the orientation regarding their immune status. There are many indicators that can be used to assess the immune system, but the greatest attention should be paid to those that change rapidly over time and reflect the body’s first line of defense. Peripheral blood granulocytes contain enzymes with strong antimicrobial properties, the level of which changes as a result of various external factors, e.g., viral infection, which was assessed in this study. The aim of the study was to evaluate the dynamics of myeloperoxidase (MPO) and lysozyme (LZM) in the experimental infection of rabbits with the *Lagovirus europaeus*/GI.1a virus, which is a pathogen causing high mortality, decimating rabbit farms all over the world in a short time. The results obtained in the dynamic system show that the levels of assessed enzymes significantly change in the blood during infection. Assessing the immune system using these indicators could therefore be a potential biomarker for the immune status of rabbits.

**Abstract:**

Infectious diseases, due to their massive scale, are the greatest pain for all rabbit breeders. Viral infections cause enormous economic losses in farms. Treating sick rabbits is very difficult and expensive, so it is very important to prevent disease by vaccinating. In order to successfully fight viral infections, it is important to know about the immune response of an infected animal. The aim of this study was to analyze the immune response mediated by antimicrobial peptides (myeloperoxidase (MPO) and lysozyme (LZM)) in peripheral blood neutrophils and rabbit serum by non-invasive immunological methods. The study was carried out on mixed breed rabbits that were experimentally infected with two strains (Erfurt and Rossi) of the *Lagovirus europaeus*/GI.1a virus. It has been observed that virus infection causes changes in the form of statistically significant increases in the activity of MPO and LZM concentration, while in the case of LZM activity only statistically significant decreases were noted. Additionally, clinical symptoms typical for the course of the disease were noted, and the probability of survival of the animals at 60 h p.i. (post infection) was 30% for the Erfurt strain, and −60% for the Rossi strain. The obtained results of MPO and LZMs suggest that these enzymes, especially MPO, may serve as a prognostic marker of the state of the immune system of rabbits.

## 1. Introduction

Rabbit breeding is an important and future-oriented direction of animal production, especially from an economic point of view. Apart from being a key element of the economy of many countries, rabbits also exist as an important link in the trophic chains of the Mediterranean ecosystems [1,2,3,4,5]. As laboratory animals, they are widely used in various types of diagnostic tests and analyses [2,6,7,8]. Currently, rabbit breeding is becoming increasingly popular. The data concerning the world production of these animals indicate that annually about 2 million of them are slaughtered exclusively for meat [9].

The loss of rabbits, especially due to various diseases, is very problematic for breeders, because even the best farms, by not following the rules of disease prevention and improper treatment, face huge problems. The treatment of sick rabbits is very difficult, as it requires cumbersome, long-lasting and systematic procedures [10,11], and the causes of rabbits’ diseases can be very different. The most common diseases of rabbits are infectious diseases, which can occur en masse, thus leading to serious losses [10,11].

One of the most dangerous diseases that can decimate rabbit breeding is rabbit haemorrhagic disease (RHD) [10,11,12]. RHD is a highly infectious and deadly viral disease manifested by acute viral hepatitis in rabbits, caused by single-stranded RNA (ssRNA) virus ssRNA virus *Lagovirus (L.) europaeus* [13,14,15]. The genus *L. europaeus* is divided into two main gene groups related to rabbit hemorrhagic disease virus (RHDV): GI or European hare syndrome virus (EBHSV): GII [14]. Within the GI gene group, four genotypes—GI.1, GI.2, GI.3, and GI.4—were distinguished. The GI.1 genotype includes classical RHDV strains (*L. europaeus*/GI.1), while the GI.2 genotype was classified as RHDV2, discovered in France in 2010 (*L. europaeus*/GI.2) [16]. Genotype GI.3 is represented by RCV-E1 (*L. europaeus*/GI.3), and GI.4 by RCV-A1 (*L. europaeus*/GI.4) and RCV-E2 (*L. europaeus*/GI.4d). The strains present in these genotypes are called nonpathogenic rabbit calicivirus (RCV). The strains of the GI.1 and GI.2 genotypes are known infectious agents responsible for the development of RHD. According to the recently proposed nomenclature [14], GI.1 is further subdivided into different antigenic variants (GI.1a-d) based on phylogeny and genetic distance [14].

Due to its features, RHD has been included in the list of notifiable diseases to the World Organisation for Animal Health (OIE). Every animal in which RHD is suspected or confirmed is immediately euthanized and cremated [10,11]. The first data of RHD were reported in Wuxi, Jiangsu Province in China in 1984 [17]. The disease was observed in the population of European rabbits (*Oryctolagus cuniculus*) of the Angora breed, which were imported from the former German Democratic Republic to China for breeding purposes [17,18]. In China, within one year, the disease contributed to the deaths of over 140 million domestic rabbits and spread over about 50,000 km^2^ [18]. The Chinese epidemic of RHD caused enormous losses in the economic sphere of the country [18,19]. The strong expansion of the virus soon led to its rapid spread throughout Europe [2]. Since 2010, the RHDV2 virus (*L. europaeus/*GI.2) has also spread to countries in Europe, Australia, America, and Africa [20]. The appearance of RHD on the Iberian Peninsula led to huge losses in the wild rabbit population [1,2,3,16,21]. There was a great interest in this problem due to the rabbit’s key contribution to the Mediterranean ecosystems, as it provides food for several endangered endemic species, such as the Iberian lynx (*Lynx pardinus*) and the Iberian imperial eagle (*Aquila heliaca*) [3]. The reduction of the wild rabbit population therefore has a negative impact on the persistence of these predators [3,22,23].

Currently, the disease is present in Europe, Asia, Africa, and Australia. Single epizooties are recorded from time to time in North America (United States, Canada, Cuba). According to the current information in the WAHID (World Animal Health Information Database) system [11], in the years 2005–2018, the occurrence of RHD was reported or suspected in 50 countries, of which more than half of the reports were reported in European countries [11]. Unfortunately, so far there is no effective cure that can save infected rabbits. The only correct method of fighting RHD is its prevention through prophylactic vaccinations [1,3,20,24].

Moreover, the key to administering proper vaccination, as well as efficient fighting against the virus, is the knowledge of the immune system response of the infected animal. In light of the above, the aim of the study was to analyze with a noninvasive immunological method, the immunological response mediated by antimicrobial peptides (myeloperoxidase and lysozyme) in neutrophils of peripheral blood and serum of rabbits infected experimentally with *L. europaeus*/GI.1a.

## 2. Materials and Methods

### 2.1. Animals

The study was conducted on a group of 25 Polish mixed-breed rabbits of various sexes, of the *Oryctolagus cuniculus* species, weighing in the range of 2.50–3.80 kg, marked as conventional animals, from licensed breeding under constant veterinary and zoo-hygienic supervision [25]. The rabbits were not vaccinated against *Lagovirus europaeus*, were in good health, and did not show any disease symptoms. Throughout the experiment, the animals stayed in a vivarium belonging to the Institute of Biology, the Faculty of Mathematical, Physical and Natural Sciences, the University of Szczecin. During the tests, appropriate zoo-technical conditions were ensured, in accordance with the recommended Polish standards developed in line with the European Union Directive with regards temperature and humidity, as well as lighting and size of cages for animals [26]. After transporting to the vivarium, the animals were subjected to a two-week adaptation period and tested for the presence of anti-RHDV antibodies using a ready-made ELISA (Institutio Zooprofilattico Sperimentale, Italy). Each of the rabbits had a separate metal cage, was fed with complete feed for rabbits (16% Rabbit, Motycz, Poland) at an amount of 0.15–0.20 kg per day, and had unlimited access to fresh drinking water. All studies were conducted with the approval of the Local Ethics Committee for Experiments on Animals in Poznań (license no. 1/2009).

### 2.2. The Scheme of the Experiment

Two antigenic variants of *Lagovirus europaeus*/GI.1a with positive hemagglutination (HA+) were selected for the study:
*L. europaeus*/GI.1a/Erfurt (Germany, 2000),*L. europaeus*/GI.1a/Rossi (Germany, 2002).

The rabbits that were used in the experiment were divided into three groups. The animals were classified into groups based on randomness. The first group (*n* = 10) were rabbits (60% female; 40% male) infected with *L. europaeus*/GI.1a/Erfurt. The second group (*n* = 10) were rabbits (50% female; 50% male) infected with *L. europaeus*/GI.1a/Rossi (*n* = 10). The third group (*n* = 5) were rabbits (40% female; 60% male) that were classified as the control group.

The virus infection occurred after the first blood sampling (0 hour of the experiment). The virus was infected by intramuscular injection of *L. europaeus*/GI.1a antigens into the lower limb muscle. Control group (*n* = 5) rabbits were given a placebo in the form of PBS (phosphate-buffered saline) in the same way. Subsequent blood samples were taken in hours: 8, 12, 24, 36, 48, 52, 56, and 60 h post infection (p.i.). Myeloperoxidase (MPO) activity was determined in peripheral blood samples, whereas in blood serum the concentration and activity of lysozyme (LZM) was determined.

### 2.3. Virus Preparation and Administration

Viruses were obtained from animals that died under natural conditions in different parts of Germany (strains were obtained in lyophilized form from Dr. Horst Schirrmeier, Friedrich Loeffler Institute Greifswald, Germany). Virus identification was performed using the real-time PCR method under the conditions previously described by Niedźwiedzka-Rystwej et al. [27]. Liver homogenate was prepared from the recovered livers, and the animals were experimentally infected in order to increase the amount of virus. After their death, the liver was prepared as a 20% homogenate, which was purified by centrifugation (3000 rpm), treatment with 10% chloroform for 60 min, and centrifugation again. Then, a suspension in glycerol was prepared in a 1:1 ratio [28,29]. All the antigens prepared in this way had the same number of virus particles, with the density from 1.310 to 1.340 g/cm^3^. Additionally, the virus titer was determined using the HA hemagglutination test, which was 1:1280 for both strains.

### 2.4. Myeloperoxidase (MPO) Activity

Determination of myeloperoxidase (MPO) activity in neutrophils was performed according to Graham’s method, as described by Zawistowski [30]. This method is based on making a smear of edetate blood, collected in a sample with an anticoagulant in the form of heparin, and the use of a color reaction with myeloperoxidase (MPO) in PMN (polymorphonuclear) cells, based on the reaction of benzidine, which in the presence of hydrogen peroxide and peroxidase passes into brown oxybenzidine. The reagent used in the color reaction with MPO was made from 250 mL of 40% ethyl alcohol, in which 350 mg of benzidine was dissolved; then the reagent was filtered and 0.35 mL of hydrogen peroxide was added. All blood smears were fixed with a mixture of formalin and ethyl alcohol, and then, for at least 3 min, were stained with a benzidine reagent. At a later stage, the preparations were stained with a 7% Giemsa solution for 7 min to stain the PMN cell nucleus. Then, the preparations prepared in this way were viewed under an optical microscope at 1000-fold magnification using immersion. In the preparations, the number of granulocytes was calculated, and the degree of color of the granules was determined according to the adopted scale (Figure 1). MPO activity was expressed as a factor, according to the intensity of the granule color in 100 PMN cells, and calculated on the basis of this formula according to Afanasyev and Kolot [31]:
(1)$$\mathrm{MPO activity ratio}=\frac{\text{Total of products of values of the degree of granule staining from}\;0\;\mathrm{to}\;4}{100\;\text{PMN cells}}\times 100$$

### 2.5. Concentration and Activity of Lysozyme (LZM)

The concentration of LZM was determined in the blood serum towards *Micrococcus (M.) lysodeikticus* using the platelet diffusion method, according to the method of Hankiewicz [32].

The substrate was a 1% agarose gel prepared from 1 g of agarose dissolved in 100 mL of 0.067 M phosphate buffer at pH 6.2, with the addition of 1 g of NaCl, in which *M. lysodeikticus* (Sigma, Saint Louis, MO, USA) was suspended (150 mg of bacteria in 15 mL phosphate buffer, at 0.067 M pH 6.2). The agarose gel prepared in this way was poured between glass plates. The agarose gel plates were allowed to set, and then 17 µL wells were carved into the gel.

The LZM standard with a concentration of 100 mg/L was obtained by dissolving 10 mg of hen egg white (Sigma, Saint Louis, MO, USA) with activity of 50,000 IU/mg in 100 mL of 0.067 M phosphate buffer at pH 6.2. A subsequent dilution at a concentration of 64 mg/L was obtained by transferring 6.4 mL of LZM standard solution (100 mg/L) into 3.6 mL of 0.067 M phosphate buffer at pH 6.2. Subsequent concentrations of 32, 16, 8, 4, 2, 1, 0.5, 0.25, and 0.125 mg/L were obtained by serial dilution by transferring 1 mL of standard from each higher concentration to 1 mL of buffer. The thus-prepared LZM standard (17 µL) and the test serum (17 µL) were spotted into the grooved wells in an agar medium, and then the plates with the medium were incubated in a humid chamber and placed in an incubator at 37 °C for 18 h. After incubation, the diameter of radiolucency zones around individual wells was measured; during this process, the size of the diameter depends on the concentration of LZM in the blood serum. The results of the LZM concentration are read from the calibration curve made on the basis of standard solutions.

In turn, the activity of LZM was calculated on the basis of the formula presented by Szmigielski [33] and presented as an indicator:
(2)$$\text{LZM activity ratio}=\frac{\text{LZM concentration in serum (mg/L)}}{{\text{absolute number of neutrophils}}^{*}}$$

The absolute neutrophil count (*) was calculated by multiplying the total leukocyte count (in 1000) by the percentage of neutrophils from the blood quality picture (leukogram).

### 2.6. Clinical Studies of Experimental Animals

Throughout the experiment, clinical symptoms and the survival of rabbits infected with *L. europaeus*/GI.1a were recorded. Rabbit survival is presented as the percentage of survival analyzed using the Kaplan–Meier method.

### 2.7. Statistical Analysis

All results were statistically analyzed using the Student’s *t*-test with Cochran–Cox correction, assuming *p* = 0.05. The analysis was performed using in the *Statistica* software, ver. 13.1 (Statsoft, Poland) and Microsoft Excel (Microsoft 365, Redmond, WA, USA).

## 3. Results

### 3.1. Myeloperoxidase (MPO) Activity

The values of MPO activity in the group of rabbits infected with *L. europaeus*/GI.1a/Erfurt ranged from 1.15 to 2.13, with the standard deviation (SD ±) within the range of 0.08–0.29 (Figure 2). On the other hand, in the case of animals infected with *L. europaeus*/GI.1a/Rossi, MPO activity ranged from 1.07 to 2.34, with the standard deviation (SD ±) from 0.16 to 0.65 (Figure 2). In the group of control rabbits, MPO activity values ranged from 1.01 to 1.27, with the standard deviation (SD ±) from 0.02 to 0.15 (Figure 2).

Both results of MPO activity in rabbits infected with *L. europaeus/*GI.1a/Erfurt and *L. europaeus*/GI.1a/Rossi throughout the entire digestion of the experiment showed a clear upward trend of the parameter under study, compared to the results of MPO activity obtained in the studies of rabbits from the control group. The obtained results were subjected to statistical analysis, which showed that both *L. europaeus*/GI.1a/Erfurt and *L. europaeus*/GI.1a/Rossi strains caused eight statistically significant changes in the form of increases in the parameter under study. These changes were observed after 8, 12, 24, 36, 48, 52, 56, and 60 h p.i. (Figure 2).

### 3.2. Lysozyme (LZM) Concentration

The parameters of LZM concentration for *L. europaeus*/GI.1a/Erfurt infected rabbits ranged from 4.22 mg/L to 6.40 mg/L, with standard deviation (SD ±) from 0.78 to 1.91 (Figure 3). On the other hand, the results recorded for *L. europaeus*/GI.1a/Rossi infection ranged from 4.03 mg/L to 5.49 mg/L, with the standard deviation (SD ±) from 0.54 to 2.12. (Figure 3). The concentration of LZM in control rabbits ranged from 3.60 mg/L to 4.50 mg/L, with the standard deviation (SD ±) ranging from 0.20 to 0.90 (Figure 3).

Both of the parameters for LZM concentration in rabbits infected with *L. europaeus*/GI.1a/Erfurt and *L. europaeus*/GI.1a/Rossi, in relation to the results obtained in the study of rabbits from the control group, showed an upward trend throughout the duration of the study. Nevertheless, they were accompanied by significant standard deviation. The obtained results were analyzed statistically, which showed that in the case of infection with *L. europaeus*/GI.1a/Erfurt, three statistically significant changes were noted, which fell at 12, 24, and 56 h p.i., while the *L. europaeus*/GI.1a/Rossi strain in within the examined parameter showed only one change of statistical significance; this change was observed at 12 h p.i. (Figure 3). High standard deviations and relatively low numbers of statistically relevant changes may be caused by the rapid losses of rabbits and the methodical character of the analysis, which depends on many outside factors.

### 3.3. Lysozyme (LZM) Activity

The results of studies on LZM activity in rabbits infected with *L. europaeus*/GI.1a/Erfurt ranged from 0.0015 to 0.0057, with the standard deviation (SD ±) from 0.0002 to 0.0047 (Figure 4). In *L. europaeus*/GI.1a/Rossi-infected animals, LZM activities ranged from 0.0004 to 0.0014, with a standard deviation (SD ±) of 0.0001 to 0.0008 (Figure 4). In the group of control rabbits, LZM activity values ranged from 0.0013 to 0.0023, with standard deviation (SD ±) from 0.0002 to 0.0008 (Figure 4).

In the case of rabbits infected with *L. europaeus*/GI.1a/Erfurt, the parameters of LZM activity, in relation to the results obtained in the study of rabbits from the control group, showed a slight upward trend at 0 hours. Then at 12 and 24 h p.i., there was a decrease in the tested parameter; from 24 to 60 h p.i., the tested parameter showed only an upward trend. Analyzing the values of LZM activity in rabbits infected with *L. europaeus*/GI.1a/Rossi, it was observed that the parameter values, in relation to the control group, showed only a downward trend throughout the duration of the experiment. The obtained result was subjected to statistical analysis, which showed that in the case of infection with *L. europaeus*/GI.1a/Erfurt, all changes within the value of the examined parameter are not statistically significant, while in the group of rabbits infected with *L. europaeus*/GI.1a/Rossi, there were five statistically significant changes in the form of decreases in the activity of LZM. These changes were observed at 8, 24, 36, 48, and 60 p.i. (Figure 4). It needs to be added that similar to LZM concentration, LZM activity is characterized by noticeably high standard deviations, which was probably caused by individual animals with relatively higher or lower results. This may lead to an overall conclusion on LZM that it is not enough stable to serve as a prognostic factor.

### 3.4. Clinical Signs Infection of Lagovirus Europaeus/GI.1a.

During the experiment, many clinical symptoms were noted, including sneezing, thickening of the blood, no response to external stimuli, rapid breathing, general body stiffness, lethargy, increased thirst, increased blood clotting, and lack of appetite. All observed clinical symptoms were characteristic of the course of RHD. The first symptoms in animals were observed as early as 8 h p.i. In the following hours of the experiment, symptoms began to worsen, and continued until the animal died or until the end of the experiment. Using the Kaplan–Meier method, we calculated the probability of survival of animals from both groups of infected rabbits and plotted them in the graph (Figure 5). In the group of animals infected with *L. europaeus*/GI.1a/Erfurt, there were seven deaths from 10 tested animals during the experiment.

The first death was recorded between 12 and 24 h p.i., then between 24 and 36 h p.i. another four deaths were recorded. The last fall of the animals was observed between 56 and 60 h after the start of the experiment. The probability of survival 60 h after infection with *L. europaeus*/GI.1a/Erfurt was 30% (Figure 5).

In the case of the group of animals infected with *L. europaeus*/GI.1a/Rossi, there were four deaths from 10 infected animals. The first two deaths, as in the case of *L. europaeus*/GI.1a/Erfurt, were recorded between 12 and 24 from the beginning of the experiment. Another two deaths were observed between 24 and 36 h p.i. The probability of survival for 60 h after infection in the group of animals infected with *L. europaeus*/GI.1a/Rossi was 60% (Figure 5).

## 4. Discussion

Results show that MPO seems to be a sensitive marker of the condition of immune system activity of infected rabbits, as it by definition increases over the course of infection. It was previously described that MPO in viral infections in increasing, and the elevated level of MPO results from the possible lysis of leukocytes, being the first line of antimicrobial defense [34,35]. After entering of the virus into the host cells, phagocytosis is performed, and the leukocytes lead to increased reactive oxygen species through MPO and NADPH oxidase, meaning that the MPO level and activity is directly correlated with the activation of phagocytes [36]. Moreover, the level of MPO enzyme is also discussed as being a prompt indicator of endothelial dysfunction, inflammation, atherosclerosis, and oxidative stress [37]. Our results show that MPO activity levels seems to be stable and behave similarly in both studied strains of *Lagovirus europaeus*/GI.1a levels of MPO—in the Erfurt strain levels were between 1.15 and 2.13, and in case of the Rossi strain the range was 1.07–2.34. The trend is similarly increasing from 8 to 60 h p.i. Also, no impact was noted and considered as significant as far as sex of the animals is concerned. Taking all of the above into consideration, it may be stated that MPO activity may be a more useful prognostic tool, compared to LZM, for defining the status of the immune system in the breed animals, in order to avoid the unexpected losses of the farmed rabbits.

Among animal models, the role of myeloperoxidase has been emphasized mainly in mice, where experiments with MPO-deficient mice showed their susceptibility to *Candida albicans* [38] and *Klebsiella pneumoniae* [39] infection. Also, in rats, infection causes decreases in MPO levels [40]. Keeping in mind that animal models are not ideally reflecting the human condition, it is worth noticing that in humans, several types of tissue injuries and pathogenesis of many diseases, like rheumatoid arthritis, cardiovascular diseases, liver disease, diabetes, and cancer, have been linked with MPO-derived oxidants [41]. Due to the above, it can be stated that elevated levels of MPO activity may be one of the best diagnostic tools of inflammatory and oxidative stress biomarkers. Finally, latest research shows that MPO may not only be pivotal for innate immunity, but seems to have a potential role in adaptive immunity [42] and autoimmune diseases [43].

Lysozyme is an enzyme hydrolyzing glycosidic linkages in bacterial peptidoglycan (PGN); due to the low toxicity of lysozyme, it is used as a natural preservative to control bacteria in meat products [44]. What is more, this enzyme may act as chitinase and activate bacterial autolysins, and it also exhibits antiviral activity against several human and animal viruses [45]. Also, since the natural substrate for lysozyme is PGN, it has similar functions as PGN recognition molecules, such as CD14, TLR2, or NOD-like proteins [46]. The antibacterial, antiviral, and antifungal roles of LZM has been claimed mainly in human-orientated studies [47]; its role in animal models is modestly represented. Yet, an extremely interesting study that corresponds to ours was found related to influenza, in which it was registered that the level of LZM is inhibited by this virus, whereas simultaneously, the virus does not influence the level of MPO [48]. So far, apart from ours, this is the only noted correlation between those potential biomarking substances of the immune system’s condition.

The mortality of the infected animals showed that the probability of survival differs in the studied viruses of *L. europaeus*/GI.1a; however, it is important to add that this condition was probably affected by several other factors impacting immune system status.

## 5. Conclusions

The results on the roles of MPO and LZM as important hallmark and prognostic factors of survival of rabbits suffering from infection with *L. europaeus*/GI.1a suggests that those antimicrobial enzymes located in neutrophils not only play a significant role in defending the host from the infection, by activating the phagocytes to actively fight with the virus, but also may serve as a prognostic marker of immune system status. Basing on the results, MPO may be a more reliable indicator of inflammatory response than LZM. Further, more extensive studies on the subject are required, especially to check the correlation between the enzymes’ activity and viral loads, in order to better understand the mechanism and use it in the protection of rabbits. Also, considering the fact that currently, *Lagovirus europaeus*/GII is widespread all around the world, it would be interesting to check if the role of MPO and LZM is repeatable.

## Figures and Tables

**Figure 1 animals-10-01581-f001:**
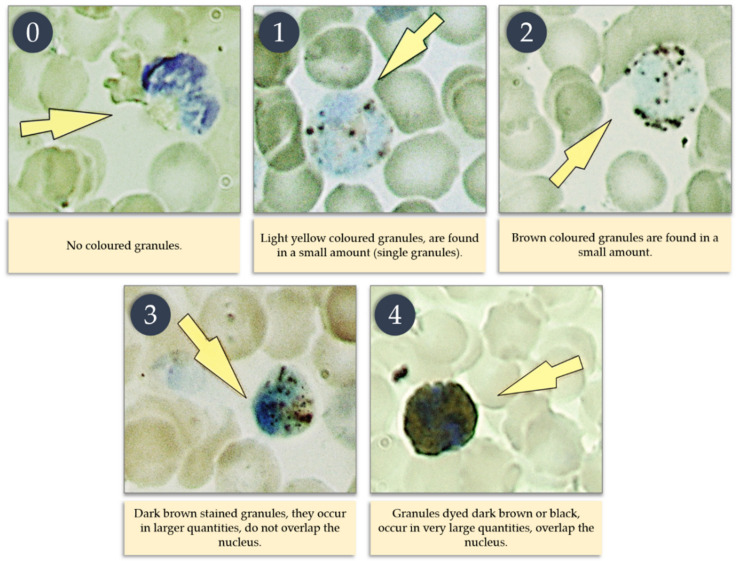
Scale showing the intensity of granule staining in PMN cells.

**Figure 2 animals-10-01581-f002:**
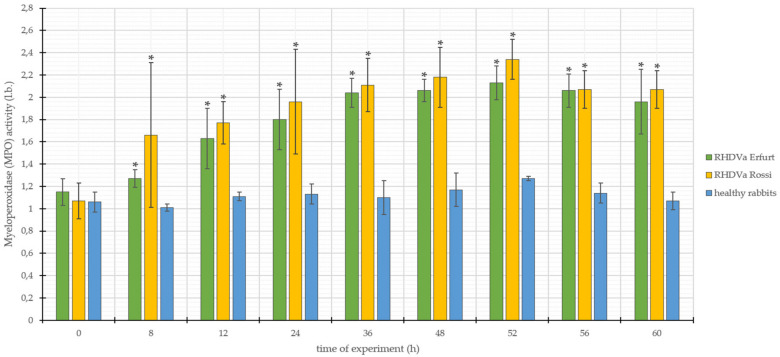
Values of myeloperoxidase (MPO) activity in rabbits infected with *L. europaeus*/GI.1a/Erfurt and *L. europaeus*/GI.1a/Rossi, as well as control rabbits. * statistically significant change with respect to the control group (*p* < 0.05).

**Figure 3 animals-10-01581-f003:**
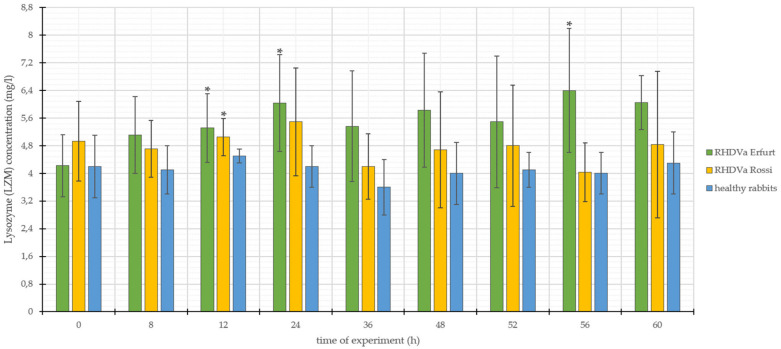
Values of lysozyme (LZM) concentration in rabbits infected with *L. europaeus*/GI.1a/Erfurt and *L. europaeus*/GI.1a/Rossi, as well as control rabbits. * statistically significant change respect to the control group (*p* < 0.05).

**Figure 4 animals-10-01581-f004:**
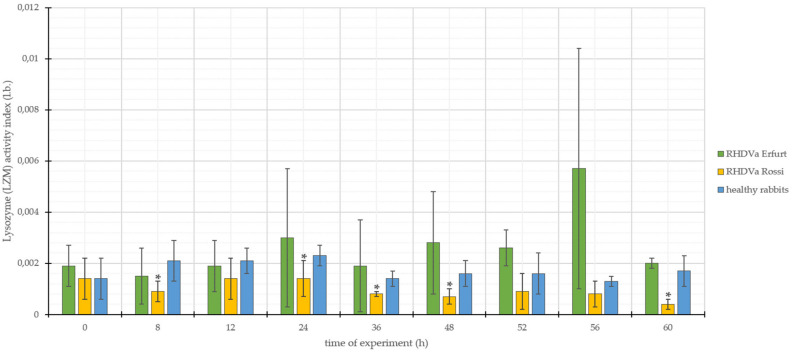
Values of lysozyme (LZM) activity in rabbits infected with *L. europaeus*/GI.1a/Erfurt and *L. europaeus*/GI.1a/Rossi, as well as control rabbits. * statistically significant change respect to the control group (*p* < 0.05).

**Figure 5 animals-10-01581-f005:**
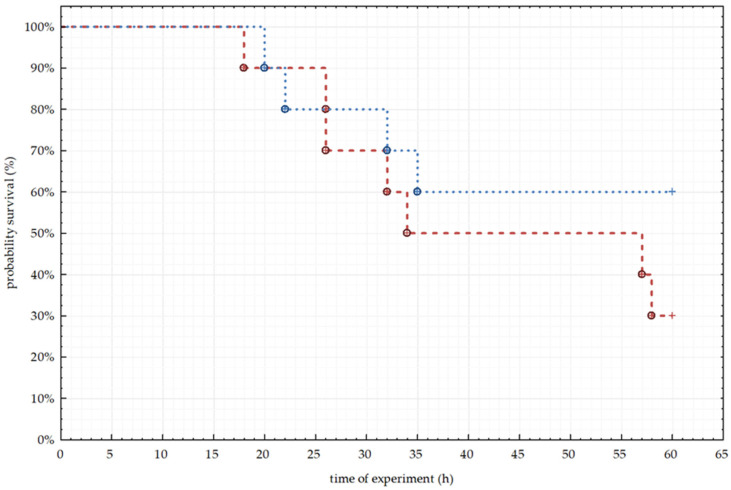
Percentage of rabbit survival recorded for the two tested strains (*L. europaeus*/GI.1a/Erfurt and *L. europaeus*/GI.1a/Rossi), analyzed by Kaplan–Meier method.

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
