# Peer review of "Myeloperoxidase and Lysozymes as a Pivotal Hallmark of Immunity Status in Rabbits"

_animals, 2020, doi:10.3390/ani10091581_

Round 1

Reviewer 1 Report

The manuscript entitled “Myeloperoxidase and lysozyme as a pivotal hallmark of immunity status in rabbits.” is important and provides a valuable information to understand the immunity status in rabbits. I think that the paper is adequate to Animals. However, the paper could be improved if the authors made the following changes:

The authors should explain that The RHDV VP60 sequences, were divided into classical RHDV G1-G5 and G6 or RHDVa (Le Gall-Reculé, 2003) and the new strain named RHDV2 (Le Gall-Recule et al, 2013). In 2017, a new RHDV nomenclature was proposed that changed the G1, G2, G3-G5 and G6, to GI.1b, GI.1c, GI.1d and GI.1a, respectively, and RHDV2 was called GI.2 (Le Pendu et al, 2017). In the fact the strains used in this work, Erfurt and Ross, work were classified in the genogroup G6 or RHDVa that currently is designed by Lagovirus europaeus GI.1a.

I also think that is important to say that in 2010, a new RHDV-related virus designated RHDV2, latter called GI.2 (Le Pendu et al, 2017), was detected, for the first time, in France in 2010 (Le Gall-Recule et al., 2013), spreading to other countries in Europe, Australia, America and Africa (Abrantes et al., 2013; Rouco et al., 2018; Lopes et al., 2019; 26 Mahar et al., 2018; Puggioni et al., 2013).

The references should be all reviewed. I will mention only five examples but there are more:

The rabbits also exist as an important link in the trophic chains of the Mediterranean ecosystems [1-5], the references 4 and 5 should be exclude, the reference 17 should be include and please add the reference Rouco et al, 2018 Transboundary emerging Disease.

As laboratory animals they are widely used in various types of diagnostic tests and analyses [2-4,6]. The references 3 and 4 should be removed please add the references Esteves et al., 2018 Exp Mol Med;  Mage et al., 2019 Dev Comp Immunol.

RHD is a highly infectious and deadly viral disease manifested by acute  viral hepatitis in rabbits, caused by an ssRNA virus Lagovirus (L.) europaeus [1-6,8,9,11,12] please maintain only the reference 12 and add the reference Vinjé et al, 2019 J Gen Virology

The first data of RHD were reported in Wuxi, Jiangsu Province in China in 1984 [1-6,8,9,13-21] please maintain only the original reference the number 13.

The disease was observed in the population of European rabbits (Oryctolagus cuniculus) of Angora breed, which were imported from the former German Democratic Republic to China for breeding purposes. Remove the sentence or maintain and keep only the references 13 and 14.

The authors must explain the differences observed between the rabbits infected with the two virus strains. The MPO activity was consistently higher in the rabbits infected with Erfurt strain and the lysozyme activity was consistently higher in the rabbits infected with Rossi strain. These two strains belong to the same genogroup, I did not expect to see differences between them. Do you expect to see big differences using for example the Lagovirus Europaeus strains GII. This is an important issue to see how reliable are these two molecules as indicators of inflammatory responses.

Author Response

Dear Reviewer,

On behalf of the authors of the article "Myeloperoxidase and lysozyme as a pivotal hallmark of immunity status in rabbits", we would like to thank you for the informative and detailed reviews of our article. We believe that your excellent knowledge and commitment influenced our article and made it much better. We followed the suggestions and tried to fulfill all suggestions, and all changes are marked in blue in the updated version of the manuscript. Here are the point-by-point answears.

Reviewer 1: 

  1. The authors should explain that The RHDV VP60 sequences, were divided into classical RHDV G1-G5 and G6 or RHDVa (Le Gall-Reculé, 2003) and the new strain named RHDV2 (Le Gall-Recule et al, 2013). In 2017, a new RHDV nomenclature was proposed that changed the G1, G2, G3-G5 and G6, to GI.1b, GI.1c, GI.1d and GI.1a, respectively, and RHDV2 was called GI.2 (Le Pendu et al, 2017). In the fact the strains used in this work, Erfurt and Ross, work were classified in the genogroup G6 or RHDVa that currently is designed by Lagovirus europaeus GI.1a.

*In line with the Reviewer's suggestions, we added a paragraph (lanes 56-66), which describes the division of RHDV into various types and antigenic variants. Additionally, we added the proposed references (Le Gall-Reculé, 2003; Le Gall-Recule et al, 2013; Le Pendu et al, 2017).

  1. I also think that is important to say that in 2010, a new RHDV-related virus designated RHDV2, latter called GI.2 (Le Pendu et al, 2017), was detected, for the first time, in France in 2010 (Le Gall-Recule et al., 2013), spreading to other countries in Europe, Australia, America and Africa (Abrantes et al., 2013; Rouco et al., 2018; Lopes et al., 2019; 26 Mahar et al., 2018; Puggioni et al., 2013).

*We have added information (lanes 60-61) about the new RHDV2 variant (year and country of discovery of RHDV2). In lanes 75-76 we added information about the spread of RHDV2 to other countries.

  1. The references should be all reviewed.

*As recommended by the Reviewer, we reviewed and corrected all references. Due to this, the numbers of references were also updated. Additionally, we followed all the suggestions of the Reviewer below.

  1. I will mention only five examples but there are more:
  • The rabbits also exist as an important link in the trophic chains of the Mediterranean ecosystems [1-5], the references 4 and 5 should be exclude, the reference 17 should be include and please add the reference Rouco et al, 2018 Transboundary emerging Disease.

*As recommended by the Reviewer, we excluded references [4] and [5], kept reference [17] and added reference (Rouco et al., 2018).

  • As laboratory animals they are widely used in various types of diagnostic tests and analyses [2-4,6]. The references 3 and 4 should be removed please add the references Esteves et al., 2018 Exp Mol Med;  Mage et al., 2019 Dev Comp Immunol.

*As recommended by the Reviewer, we removed references [3] and [4] and added references Esteves et al., 2018, Exp Mol Med and Mage et al., 2019 Dev Comp Immunol.

  • RHD is a highly infectious and deadly viral disease manifested by acute  viral hepatitis in rabbits, caused by an ssRNA virus Lagovirus (L.) europaeus [1-6,8,9,11,12] please maintain only the reference 12 and add the reference Vinjé et al, 2019 J Gen Virology.

*As recommended by the Reviewer, we have removed the references [1] [2] [3] [4] [5] [6] [8] [9] [11] and we only kept the reference [12]. Additionally, as suggested by the Reviewer, reference has been added (Vinjé et al., 2019 J. Gen. Virology.).

  • The first data of RHD were reported in Wuxi, Jiangsu Province in China in 1984 [1-6,8,9,13-21] please maintain only the original reference the number 13.

*As suggested by the Reviewer, we have kept only the original reference. Other references [1] [2] [3] [4] [5] [6] [8] [9] [13] [14] [15] [16] [17] [18] [19] [20] [21] we have removed.

  • The disease was observed in the population of European rabbits (Oryctolagus cuniculus) of Angora breed, which were imported from the former German Democratic Republic to China for breeding purposes. Remove the sentence or maintain and keep only the references 13 and 14.

*We have kept the sentence "The disease was observed in the population of European rabbits (Oryctolagus cuniculus) of Angora breed, which were imported from the former German Democratic Republic to China for breeding purposes". As recommended by the Reviewer, we only kept the references [13] and [14].

  1. The authors must explain the differences observed between the rabbits infected with the two virus strains. The MPO activity was consistently higher in the rabbits infected with Erfurt strain and the lysozyme activity was consistently higher in the rabbits infected with Rossi strain. These two strains belong to the same genogroup, I did not expect to see differences between them. Do you expect to see big differences using for example the Lagovirus Europaeus strains GII. This is an important issue to see how reliable are these two molecules as indicators of inflammatory responses.

  • After analyzing the results again, we added information on MPO levels in discussion, as we think that MPO may be more stable comparing to LZM and may suit better as an indicator. In LZM there are several differences noted in levels and activity, and since the strains, as noted by the Reviewer, are in the same genogroup, we should be more careful in pointing it as a biomarker. Since now, we have no data on the immunological status of these two molecules in rabbits infected with Lagovirus europaeus GII and it would be very interesting to check it. This was also added as a conclusion.

Thank you for your time and consideration, 

Paulina Niedźwiedzka-Rystwej 

Reviewer 2 Report

In the present manuscript, Hrynkiewicz et al. have analyzed the dynamics of antimicrobial enzymes in peripheral blood neutrophils and rabbit serum in the course of Lagovirus europaeus GI.1a infection. They have observed an increase of MPO activity and LZM concentration in infected animals and a decrease in LZM activity. They conclude that these enzymes can serve as a prognostic marker of immune system status of rabbits.

The introduction and methods are clear and well written. The results section is clear and well organized but needs to be revised concerning the descriptions of their results – see details below. The discussion and conclusion are accurately carried out but a critical discussion concerning LZM needs to be added. Overall, this is a clear and nicely presented work.

Lanes 192-195: there is obviously a mistake concerning the MPO activity given here, the range for the Rossi strain is probably rather about 1.05 to 2.3 and not the same as for the control group.

Lanes 196-198: the upward trend is not lasting throughout the entire duration of the experiment, there is a drop in MPO activity after 52h of infection for both strains. Or, do they mean an elevated level for infected rabbits in comparison to non-infected animals throughout the experiment, so they should rephrase this sentence to make it clearer.

Lanes 198-201: It is not clear what is meant with statistically significant change here, is the change infected to non-infected animal meant or from one time point compared to the other. The time points from 12 to 60 h p.i. are given here but “*” for statistically significant change is also given for the 8h time point in the diagram. Please rephrase this section to make it more plausible.

Lanes 214-220: The authors should evaluate their results here more critically, as they have huge standard deviations and only a few statistically relevant changes.

Section 3.3: Like for LZM concentration the authors should be more critical concerning their conclusions here – upward and downward trends - considering huge standard deviations and few statistically relevant changes at least for the Erfurt strain.

Discussion: The authors should discuss their results concerning the LZM concentration and activity, the huge standard deviations and the lack of statistically relevant data. They should discuss if this enzyme is suitable as a prognostic factor.

Author Response

Dear Reviewer,

On behalf of the authors of the article "Myeloperoxidase and lysozyme as a pivotal hallmark of immunity status in rabbits", we would like to thank you for the informative and detailed reviews of our article. We believe that your excellent knowledge and commitment oinfluenced our article and made it much better. We followed the suggestions and tried to fulfill all suggestions, and all changes are marked in green in the updated version of the manuscript. Here are the point-by-point answears.

Reviewer 2: 

  1. Lanes 192-195: there is obviously a mistake concerning the MPO activity given here, the range for the Rossi strain is probably rather about 1.05 to 2.3 and not the same as for the control group.

*We improved the range of parameters and the standard deviation of MPO activity for the Rossi strain. The valid range is 1.07 - 2.34 with a standard deviation (SD ±) of 0.16 to 0.65. We apologize for the mistake. This shouldn't have happened.

  1. Lanes 196-198: the upward trend is not lasting throughout the entire duration of the experiment, there is a drop in MPO activity after 52h of infection for both strains. Or, do they mean an elevated level for infected rabbits in comparison to non-infected animals throughout the experiment, so they should rephrase this sentence to make it clearer.

*We reformulated the sentence to (hopefully) add the missing logic. In the given paragraph, we meant the increase of the MPO activity parameter in infected rabbits compared to the MPO activity parameter in rabbits from the control group.

  1. Lanes 198-201: It is not clear what is meant with statistically significant change here, is the change infected to non-infected animal meant or from one time point compared to the other. The time points from 12 to 60 h p.i. are given here but “*” for statistically significant change is also given for the 8h time point in the diagram. Please rephrase this section to make it more plausible.

*We reformulated and corrected (lanes 208-210) the sentence. We changed from "seven statistically significant changes" to "eight statistically significant changes". Additionally, we added the missing parameter increase at 8 p.i. We sincerely apologize for the mistake.

  1. Lanes 214-220: The authors should evaluate their results here more critically, as they have huge standard deviations and only a few statistically relevant changes.
  2. Section 3.3: Like for LZM concentration the authors should be more critical concerning their conclusions here – upward and downward trends - considering huge standard deviations and few statistically relevant changes at least for the Erfurt strain.
  3. Discussion: The authors should discuss their results concerning the LZM concentration and activity, the huge standard deviations and the lack of statistically relevant data. They should discuss if this enzyme is suitable as a prognostic factor.

*After analyzing the results again, and added critical discussion to the LZM concentration and activity, but this also led to a change in concluding, as we now clearly see that in comparison to MPO, LZM may not be stable and relevant enough to serve as a prognostic factor. This was reconstructed in the manuscript.

Thank you for your time and consideration,

Paulina Niedźwiedzka-Rystwej

Reviewer 3 Report

The report by Hrynkiewicz and colleagues evaluates the reliability of two blood immune molecules as biomarkers of infection in rabbits. The findings of this study could be translated into a monitoring tool for infectious diseases in industrial lifestock production.

Contrary to the authors’ statement, I do not think authors have evidence to support the use of lysozyme as a pivotal hallmark of immunity status. It looks like that lysozyme values trended to be higher in animals infected with the less virulent strain (Erfurt). Authors should evaluate if this is the case (e.g. correlating the individual values for this marker with the number of days that the corresponding animal survived).

Methodological considerations:

  • Animals were divided in three experimental groups. Was this through a randomization process? If yes, please state in the text. Please also confirm using appropriate statistical tests that animals from different sex/race were equally distributed across study groups.
  • Authors used rabbits from different race and sex. It must be confirm that these variables were not interacting with the experimental outcomes.
  • The probability of survival shows a marked difference between the viruses studied. So far, this fact is limited in the discussion to lines 306-308, and in my eyes the argument is pretty vague. Authors should demonstrate that the integrity of the viral inoculum was not different. GI.1a Erfurt viruses may be attenuated via the purification process. Is there any evidence of differences in virulence between Erfurt and Rossi viral strains?
  • How did the authors ensure the integrity of the viral particles? How did they ensure that the animals were infected with the same number of competent viral particles?

Minor comments:

  • Can the authors clarify what they mean with “country of origin” in (lines 121-122): “Viruses were obtained from animals that died under natural conditions and were found in the 121 country of origin”. Were these viruses obtained from different countries? If yes, this should be detailed in the corresponding section of the material and methods.

Author Response

Dear Reviewer,

On behalf of the authors of the article "Myeloperoxidase and lysozyme as a pivotal hallmark of immunity status in rabbits", we would like to thank you for the informative and detailed reviews of our article. We believe that your excellent knowledge and commitment influenced our article and made it much better. We followed the suggestions and tried to fulfill all suggestions, and all changes are marked in grey in the updated version of the manuscript. Here are the point-by-point answears.

Reviewer 3: 

  1. Animals were divided in three experimental groups. Was this through a randomization process? If yes, please state in the text. Please also confirm using appropriate statistical tests that animals from different sex/race were equally distributed across study groups.
  • Details were added in the text (section 2.2).
  1. Authors used rabbits from different race and sex. It must be confirm that these variables were not interacting with the experimental outcomes.
  • Details of breed and sex was added in the text (section 2.1). We assumed, basing on our previous studies (Tokarz-DeptuÅ‚a et al. Pol. J. Vet. Sci. 18, 1928, 2015) that there is no impact of sex on the studied parameters.
  1. The probability of survival shows a marked difference between the viruses studied. So far, this fact is limited in the discussion to lines 306-308, and in my eyes the argument is pretty vague. Authors should demonstrate that the integrity of the viral inoculum was not different. GI.1a Erfurt viruses may be attenuated via the purification process. Is there any evidence of differences in virulence between Erfurt and Rossi viral strains?
  • Unfortunately, no studies were performed to evidence the difference or similarity on the virulence of the studies strains, yet we have performed real-time PCR test after the deaths of the infected animals and it was showed that both strains were RHDV-positive, with Tm 87,28 and Tm 87,51, and Ct 12 and 13, respectively. We did not included this in the text of the manuscript.
  1. How did the authors ensure the integrity of the viral particles? How did they ensure that the animals were infected with the same number of competent viral particles?
  • The antigens were prepared identically (as in section 2.3) and the density of viral particles infecting the rabbits were estimated to vary between 1.310 and 1.340 g/cm3.
  1. Can the authors clarify what they mean with “country of origin” in (lines 121-122): “Viruses were obtained from animals that died under natural conditions and were found in the 121 country of origin”. Were these viruses obtained from different countries? If yes, this should be detailed in the corresponding section of the material and methods.

*We have added information about the country of origin of the RHDVa strains.

Thank you for your time and consideration, 

Paulina Niedźwiedzka-Rystwej 

Round 2

Reviewer 3 Report

The authors have reasonably addressed my questions.

Please check typos in the main text.

Author Response

Dear Reviewer,

Thank you for your answer. We tried to correct typos in the text.

Regards,

Paulina Niedźwiedzka-Rystwej